# GRADIENT DESCENT RESISTS COMPOSITIONALITY

## ABSTRACT

In this paper, we argue that gradient descent is one of the reasons that make compositionality learning hard during neural network optimization. We find that the optimization process imposes a bias toward non-compositional solutions. This is caused by gradient descent, trying to use all available and redundant information from input, violating the conditional independence property of compositionality. Based on this finding, we suggest that compositionality learning approaches considering only model architecture design are unlikely to achieve complete compositionality. This is the first work to investigate the relation between compositional learning and gradient descent. We hope this study provides novel insights into compositional generalization, and forms a basis for new research directions to equip machine learning models with such skills for human-level intelligence. The source code is included in supplementary material.

## 1 INTRODUCTION

Compositional generalization is the algebraic capacity to understand and produce many novel combinations from known components (Chomsky, 1957; Montague, 1970), and it is a key element of human intelligence (Minsky, 1986; Lake et al., 2017) to recognize the world efficiently and create imagination. Broadly speaking, compositional generalization is a class of out-of-distribution generalization (Bengio, 2017), where the training and test distributions are different. A sample in such a setting is a combination of several components, and the generalization is enabled by recombining the seen components of the unseen combination during inference. For example, in the image domain, an object is a combination of multiple parts or properties. In the language domain, a sentence is a combination of syntax and semantics. Each component of an output depends only on the corresponding input component, but not on other variables. We call this the *conditional independence property*, and will formally introduce in Section 3.

People hope to design machine learning algorithms with compositional generalization skills. However, conventional neural network models generally lack such ability. There have been many attempts to equip models with compositionality (Fodor & Pylyshyn, 1988; Bahdanau et al., 2019), and most efforts focus on designing neural network architectures (Graves et al., 2014; Andreas et al., 2016; Henaff et al., 2016; Shazeer et al., 2017; Li et al., 2018; Santoro et al., 2018; Kirsch et al., 2018; Rosenbaum et al., 2019; Goyal et al., 2019). Recently, multiple approaches showed progress in specific tasks (Li et al., 2019; 2020; Lake, 2019; Russin et al., 2019), but we still do not know why standard approaches seldom achieve good compositionality in general.

In this paper, we argue that there is a bias to prevent parameters from reaching compositional solutions, when we use gradient descent in optimization (please see Figure 1 for illustrations). This is because gradient seeks the steepest direction, so that it uses all possible and redundant input information, which contradicts to the conditional independence property of compositionality. This problem is not due to how gradient is computed, such as back propagation, but caused by the essential property of gradient. We derive theoretical relation between gradient descent and compositionality with information theory. We also provide examples and visualization to show the detailed process of how gradient resists compositionality. Based on the finding, we propose that compositionality learning approaches with model structure design (manual or searching) alone are not likely to achieve complete compositionality.

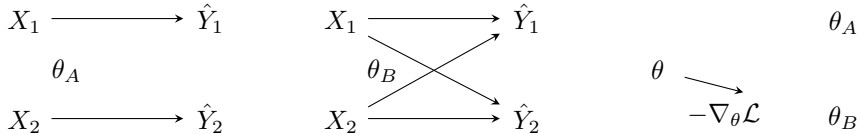

Figure 1: Conceptual illustration of compositionality and the impact of gradient descent. $X_1, X_2$ are entangled input, and $\hat{Y}_1, \hat{Y}_2$ are entangled output. $\hat{Y}_i$ aligns with $X_i$, for $i = 1, 2$. (Left) Compositional solution with $\theta_A$. (Middle) Non-compositional solution with $\theta_B$. (Right) In parameter space, gradient descent encourages parameters closer to $\theta_B$, than $\theta_A$, hence resisting compositionality.

We hope this research provides new insights and forms a basis for new research directions in compositional generalization, and helps to improve machine intelligence towards human-level. The contributions of this paper can be summarized as follows.

- The novelty of this work is to find the relation between compositional learning and gradient descent in optimization process, i.e., gradient descent resists compositionality.

- We theoretically derive the result and explain why standard approaches with architecture design alone do not address compositionality.

## 2 RELATED WORK

**Compositionality**    Humans learn language and recognize the world in a flexible way by leveraging *systematic compositionality*. The compositional generalization is critical in human cognition (Minsky, 1986; Lake et al., 2017), and it helps humans to connect limited amount of learned concepts for unseen combinations. Though deep learning has many achievements in recent years (LeCun et al., 2015; Krizhevsky et al., 2012; Yu & Deng, 2012; He et al., 2016; Wu & et al, 2016), compositional generalization has not been well addressed (Fodor & Pylyshyn, 1988; Marcus, 1998; Fodor & Lepore, 2002; Marcus, 2003; Calvo & Symons, 2014).

There are observations that current neural network models do not learn compositionality (Bahdanau et al., 2019). Most recently, multiple approaches are proposed to address compositionality in neural networks (Li et al., 2019; 2020; Lake, 2019; Russin et al., 2019) for specific tasks. However, we are still not sure why compositionality is hard to achieve in general cases, and this work discusses about this problem from optimization perspective.

Another line of related work is independent disentangled representation learning (Higgins et al., 2017; Locatello et al., 2019). Its main assumption is that the expected components are statistically independent in training data. This setting does not have transferring problem in test, because all combinations have positive joint probabilities in training (please refer to Section 3).

Compositionality is applied in different areas such as continual learning (Jin et al., 2020; Li et al., 2020), question answering (Andreas et al., 2016; Hudson & Manning, 2019; Keysers et al., 2020), and reasoning (Talmor et al., 2020).

**Gradient Descent**    Gradient descent is a powerful and general purpose optimization tool for solving large scale problems in deep neural networks. It is usually used in a stochastic way (Stochastic Gradient Descent) with mini-batches, and has many variations such as Momentum (Rumelhart et al., 1986), averaging (Polyak & Juditsky, 1992), AdaGrad (Duchi et al., 2011), AdaDelta (Zeiler, 2012), RMSProp (Tieleman & Hinton, 2012), Adam (Kingma & Ba, 2014).

Most of the previous work focus on faster reduction of loss and theoretical convergence analysis of SGD (Bottou et al., 2018; Luo, 1991; Reddi et al., 2018; Chen et al., 2018; Zhou et al., 2018; Zou & Shen, 2018; De et al., 2018; Zou et al., 2018; Ward et al., 2018; Barakat & Bianchi, 2019). In particular, this work focuses on investigating why standard neural network training only achieves limited level of compositionality by studying the relationship between gradient descent and compositionality.

# 3 CONCEPTS FOR COMPOSITIONALITY AND GRADIENT DESCENT

We first formulate compositionality using the conditional independence property, and define compositional generalization. We then review properties of gradient for the derivation in the next section.

**Conditional Independence Property for Compositionality**   When multiple hidden variables live in the same representation, and cannot be separated by simply splitting the representation, then these variables are entangled in the representation. For example, color and shape are two hidden variables and they share the same representation of image. Also, syntax and semantics are two hidden variables and they share the same representation of sentence. When we extract the hidden variables from their shared representation, we disentangle them.

We then consider a prediction problem, where input $X$ and output $Y$ have multiple entangled components that are not labeled in data, and they are aligned. For example in machine translation, $X_1$ is input syntax, and $X_2$ is input semantics. $Y_1$ is output syntax, and $Y_2$ is output semantics. The syntax of output $Y_1$ depends only on the syntax of input $X_1$, and the semantics of output $Y_2$ depends only on the semantics of $X_2$. We can formalize the alignments as *conditional independence property*: $Y_i$ depends only on $X_i$.

$$\forall i : P(Y_i|X_1,\ldots,X_K,Y_1,\ldots,Y_{i-1},Y_{i+1},\ldots,Y_K) = P(Y_i|X_i).$$

When a model fits this property, we say it has *compositionality*. Note that this can be understood as a kind of sparseness property (Bengio, 2017), because it restricts effective connection between input and output components.

**Compositional Generalization**   In compositional generalization, each sample in either training or test is a combination of several components. A test sample has a combination that does not appear in training, but each component of the test sample appears in training. We need to recombine the seen components to generalize to the test sample. We can define *compositional generalization* probabilistically as follows.

In train,
$$\forall i : P(X_i) > 0, P(X_1,\ldots,X_K) = 0,$$
$$\forall i : P(Y_i|X_i) \text{ is high.}$$

In test,
$$P(X_1,\ldots,X_K) > 0,$$
$$P(Y_1,\ldots,Y_K|X_1,\ldots,X_K) \text{ is predicted high.}$$

Compositionality bridges the gap between training and test distributions to achieve compositional generalization. We first apply chain rule, and then use compositionality as follows.

$$P(Y_1,\ldots,Y_K|X_1,\ldots,X_K) = \prod_{i=1}^{K} P(Y_i|X_1,\ldots,X_K,Y_1,\ldots,Y_{i-1}) = \prod_{i=1}^{K} P(Y_i|X_i).$$

When $P(Y_i|X_i)$ are all high, their product should also be high. Therefore, a model with compositionality—satisfying this conditional independence property—addresses compositional generalization.

**Property of Gradient**   For a function $f(x_1,\ldots,x_K)$, the gradient $\nabla f$ is the steepest direction to change the function's value. Generally, gradient descent methods estimate $\nabla f$ using low-order local estimation. By definition, it is the vector of partial derivatives with respective to the inputs.

$$\nabla f = \frac{\partial f}{\partial x_1},\ldots,\frac{\partial f}{\partial x_K}$$

We will use the following definition in later arguments.

**Definition 1** (Partial derivative). *Partial derivative for an input is the derivative assuming other inputs are constant.*

$$\forall i = 1,\ldots,K : \frac{\partial f(x_1,\ldots,x_K)}{\partial x_i} = \frac{df(c_1,\ldots,x_i,\ldots,c_K)}{dx_i}$$

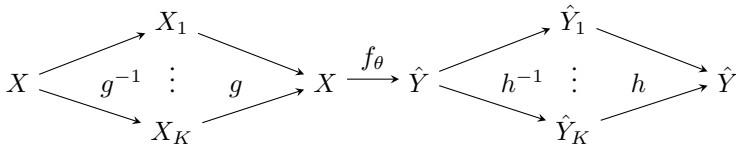

Figure 2: Extended neural network structure. Middle part is original model structure (one input and one output). Extending with $X, X_1, \ldots, X_K$ (left) corresponds to entangled input. Extending with $\hat{Y}, \hat{Y}_1, \ldots, \hat{Y}_K$ (right) corresponds to entangled output.

## 4    GRADIENT DESCENT RESISTS COMPOSITIONALITY

We focus on the early phase of training to show that gradient descent causes a model to use the redundant information to compute output when it has information to reduce the loss. We develop the arguments step by step. We first analyze the influence of the input on an output variable. We then consider the case of entangled inputs and one output. Finally, we discuss the case with entangled inputs and entangled outputs.

The gradient is used to reduce loss, so we aim to relate loss reduction and the influence from input to output. To do that, we use mutual information to describe the influence, and use knowledge from information theory (Definition 2, Theorem 1 and Theorem 2). We also study the impact of gradient descent to the influence, so we relate mutual information with gradient with Proposition 1.

**Definition 2** (Markov chain (Cover, 1999) p.34). *Random variables X, Y, Z are said to form a Markov chain in that order (denoted by $X \to Y \to Z$) if the conditional distribution of Z depends only on Y and is conditionally independent of X.*

**Theorem 1** (Data-processing inequality (Cover, 1999) p.34). *If $X \to Y \to Z$, then $I(X;Y) \geq I(X;Z)$.*

**Theorem 2** (Chain rule for information (Cover, 1999) p.24). *For random variables $X, Y, Z$, $I(X,Y;Z) = I(Y;Z|X) + I(X;Z)$.*

**Proposition 1.** *When $\frac{\partial Y}{\partial X}$ is defined, $I(X;Y) > 0 \iff \frac{\partial Y}{\partial X} \neq 0$*

*Proof.* $I(X;Y) > 0$ means $X$ and $Y$ are not independent, which means $Y$ is not invariant to $X$. $\quad\square$

### 4.1    ONE INPUT AND ONE OUTPUT

We first consider a basic setting that the data has a single input $X$ and output $Y$. A model $f$ with parameters $\theta$ has input $X$ and output $\hat{Y}$ (Figure 2 middle). We optimize a loss function $\mathcal{L}$. Applying the gradient $\nabla_\theta \mathcal{L}(Y, \hat{Y})$ reduces $\mathcal{L}(Y, \hat{Y})$, bringing $Y$ and $\hat{Y}$ closer. Since $Y$ changes according to $X$, $\hat{Y}$ is encouraged to change according to $X$. We look into details as follows.

In the common supervised learning setting, given $X$, the ground truth $Y$ does not depend on prediction $\hat{Y}$, which means they form a Markov chain $\hat{Y} \to X \to Y$. We do not require specific form of the loss function, $\mathcal{L}$, but we assume that when it is reduced, $\hat{Y}$ moves closer to $Y$, and increases the mutual information $I(\hat{Y}, Y)$. Many widely used loss functions encourage increased mutual information between model output and dataset labels. Also, training algorithms are designed to reduce loss when the input has information to do so. These assumptions are likely to hold especially in the early part of training. We also use local linear approximation when discussing gradients, i.e. $dx = \Delta x$. We derive the proof by studying relations between random variables, and show that gradient descent increases the lower bound of mutual information between model input and output, hence the output is dependent on the input.

**Proposition 2.** *If a small change of parameters $\Delta\theta$ increases $I(Y; \hat{Y}_\theta)$, then $I(X; \hat{Y}_{\theta+\Delta\theta})$ is positive with parameters $\theta + \Delta\theta$. $\forall\Delta\theta : \Delta I(Y; \hat{Y}_\theta) > 0 \implies I(X; \hat{Y}_{\theta+\Delta\theta}) > 0$.*

*Proof.* Since $Y$ and $\hat{Y}$ are conditionally independent given $X$, i.e., $\hat{Y} \to X \to Y$, with data-processing inequality (Theorem 1), $I(\hat{Y}; X) \geq I(\hat{Y}; Y)$. So $I(Y; \hat{Y}_{\theta+\Delta\theta})$ is a lower bound of $I(X; \hat{Y}_{\theta+\Delta\theta})$. By definition, $\Delta I(Y; \hat{Y}_\theta) = I(Y; \hat{Y}_{\theta+\Delta\theta}) - I(Y; \hat{Y}_\theta)$, so that $I(Y; \hat{Y}_{\theta+\Delta\theta}) = I(Y; \hat{Y}_\theta) + \Delta I(Y; \hat{Y}_\theta)$. We also have $I(Y; \hat{Y}_\theta) \geq 0$ by definition. Therefore $I(Y; \hat{Y}_{\theta+\Delta\theta}) > 0$.

$$I(X; \hat{Y}_{\theta+\Delta\theta}) \geq I(Y; \hat{Y}_{\theta+\Delta\theta}) = I(Y; \hat{Y}_\theta) + \Delta I(Y; \hat{Y}_\theta) > 0$$

$\square$

**Proposition 3.** *If $X$ has information to reduce loss $\mathcal{L}(Y, \hat{Y})$, $\frac{\partial \hat{Y}}{\partial X} \neq 0$ for updated parameters.*

*Proof.* Since training algorithm reduces loss, with local linear approximation, gradient descent reduces loss. This increases mutual information, so that Proposition 2 applies. So we have $I(X; \hat{Y}_{\theta+\Delta\theta}) > 0$. With Proposition 1, we have $\frac{\partial \hat{Y}}{\partial X} \neq 0$ for the updated parameters $\theta + \Delta\theta$. $\square$

### 4.2 Entangled Input and One Output

Then, we study the case where input $X$ is entanglement of multiple hidden input components $X_1, \ldots, X_K$, and output is a single variable $\hat{Y}$ that depends only on $X_i$. We hope to make $\hat{Y}$ invariant to $X_j, \forall j \neq i$. For example, in a parsing task (Li & Eisner, 2019), output parse tree $Y$ depends only on the input syntax $X_1$, but not on input semantics $X_2$.

For the convenience of analysis, we assume we have a fixed differentiable oracle encoder network $g$ and decoder network $g^{-1}$. $g^{-1}$ maps $X$ to $X_1, \ldots, X_K$, and $g$ maps them back.

$$X = g(X_1, \ldots, X_K) \qquad\qquad X_1, \ldots, X_K = g^{-1}(X)$$

We extend the model structure with $g$ and $g^{-1}$ and use the input to $g^{-1}$ as model input (Figure 2 left and middle).

$$\hat{Y} = f_\theta \circ g \circ g^{-1}(X)$$

This model is exactly the same as the original one, because there is no additional trainable parameters, and $g \circ g^{-1}$ does not change $X$.

**Proposition 4.** *If $X_i$ has information to reduce loss $\mathcal{L}(Y, \hat{Y})$, $\frac{\partial \hat{Y}}{\partial X_i} \neq 0$ for updated parameters.*

*Proof.* With the property of gradient (Definition 1), we can regard $X_j, \forall j \neq i$ as constant values when computing the gradient w.r.t. $X_i$. So, with linear approximation, Proposition 3 applies. $\square$

### 4.3 Entangled Input and Entangled Output

We then discuss the case that output $Y$ is also the entanglement of $Y_1, \ldots, Y_K$. $Y_i$ depends only on $X_i$ for all $i = 1, \ldots, K$. This corresponds to the example of machine translation.

We assume we have a fixed differentiable oracle encoder network $h$ and decoder network $h^{-1}$. $h$ takes $\hat{Y}$ as input and produce $K$ outputs $\hat{Y}_1, \ldots, \hat{Y}_K = h(\hat{Y})$, and $h$ maps them back. We extend the model structure with $h$ and $h^{-1}$ (Figure 2).

$$\hat{Y} = h \circ h^{-1} \circ f_\theta \circ g \circ g^{-1}(X)$$

This model is the same as the original one, because there is no additional trainable parameters and $h \circ h^{-1}$ does not change $\hat{Y}$. We derive proof with this extended model. The intuitive idea is that the reduction of loss will make each $Y_i$ contain more information of $Y$, and for each $Y_i$, we can apply previous discussion. We denote $\hat{Y}_{\neq i} = \hat{Y}_1, \ldots, \hat{Y}_{i-1}, \hat{Y}_{i+1}, \ldots, \hat{Y}_K$.

**Proposition 5.** *If $X_j$ has information to reduce loss $\mathcal{L}(Y, \hat{Y})$ through the change of $\hat{Y}_i$, then $\frac{\partial \hat{Y}_i}{\partial X_j} \neq 0, \forall j \neq i$ for updated parameters.*

*Proof.* We first use chain rule of gradient to separate the gradient to sum of $K$ terms, each corresponding to a $\hat{Y}_i$: $\frac{\partial \mathcal{L}}{\partial \theta} = \sum_{i=1}^{K} \frac{\partial \hat{Y}_i}{\partial \theta} \frac{\partial \mathcal{L}}{\partial \hat{Y}_i}$. We look at a term for $\frac{\partial \hat{Y}_i}{\partial \theta} \frac{\partial \mathcal{L}}{\partial \hat{Y}_i}$. $\hat{Y}_{\neq i}$ are constant when computing

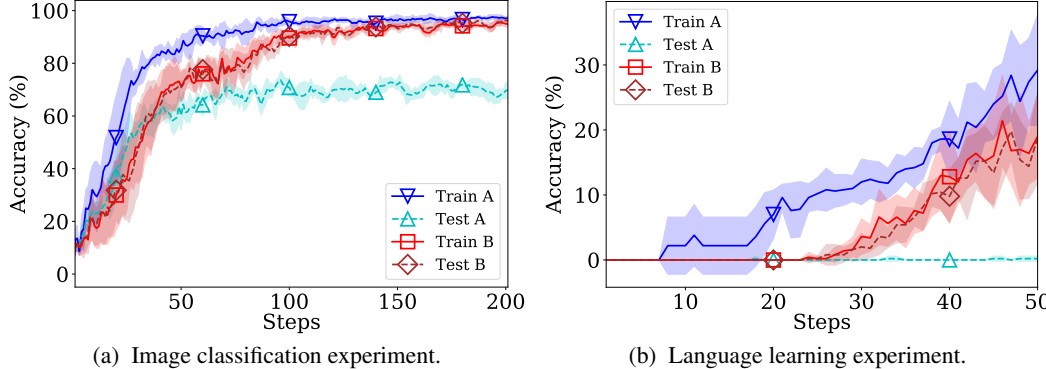

(a) Image classification experiment.        (b) Language learning experiment.

Figure 3: Results for both the first (Train/Test A) and second (Train/Test B) settings. In the first setting, the training performance increases rapidly (blue), but the test performance (cyan) is not close to the training one. In the second setting, the training (red) and test (brown) performances are close. This means that the gradient descent uses the second input to accelerate training, but it lacks compositionality.

$\frac{\partial \mathcal{L}}{\partial \hat{Y}_i}$ (Definition 1), and $\hat{Y}_{\neq i}$ are not included in $\frac{\partial \hat{Y}_i}{\partial \theta}$, so that we can treat $\hat{Y}_{\neq i}$ as constant for this term. We then look at the conditional mutual information $I(\hat{Y}_i; Y | \hat{Y}_{\neq i})$ for this term. With chain rule for information (Theorem 2), we have $I(\hat{Y}; Y) = I(\hat{Y}_i; Y | \hat{Y}_{\neq i}) + I(\hat{Y}_{\neq i}; Y)$. Since $Y$ and $\hat{Y}_{\neq i}$ are both fixed, $I(\hat{Y}_{\neq i}; Y)$ is fixed. So the change of $I(\hat{Y}; Y)$ equals to the change of $I(\hat{Y}_i; Y | \hat{Y}_{\neq i})$. Therefore, the reduction of loss $\mathcal{L}$ increases the mutual information for the component. We can then apply Proposition 4. With linear approximation, the change of parameters ($\Delta\theta$) computed with gradient descent is the sum of changes of parameters ($\Delta\theta_i$) for each term, i.e., $\Delta\theta = \sum_{i=1}^{K} \Delta\theta_i$. It is unlikely that a $\Delta\theta_i$ can be cancelled by the sum of other $\Delta\theta_j, j \neq i$, because each component corresponds to different underlying factors, and they are not contradictory. Hence, this proposition holds.  □

In Proposition 4 and Proposition 5, the gradient is not zero, so an output depends on redundant input (Proposition 2) in both cases. Therefore, gradient descent resists conditional independence property of compositionality.

## 5   EXAMPLES

In this section, we show example cases to emphasize that the theoretical result occurs practically. We focus on the conditional independence property, and we design the experiments in the following way. To test the compositionality, we use different training and test distributions, and the test prediction requires compositional generalization. We measure training and test accuracy to evaluate the ability for compositional generalization.

We use two settings in each experiment. In the first setting (A), we use both $X_1$ and $X_2$ as input to the model. In the second setting (B), we only use $X_1$, and remove information of $X_2$ by setting it to be a random input. The model architecture and other settings are the same. By comparing the test performance in the two settings, we show that a model can be trained faster with $X_2$, but it does not hold compositionality. We run experiments for 5 times, and plot the mean and variance at each training step.

### 5.1   IMAGE CLASSIFICATION

We use MNIST dataset (LeCun et al., 1998) in this experiment. The dataset contains pairs of input image and output label. An image is gray scale with fixed size, and a label is in the set of ten possible values $\{0, 1, \ldots, 9\}$. We use two original samples $(X_1, Y_1), (X_2, Y_2)$ to make one sample $(X, Y)$. $X$ is the horizontal concatenation of $X_1$ and $X_2$, and $Y = Y_1$. Note that a generated sample does not directly use the label of the second sample $Y_2$.

| **jump** | JUMP |
| walk before run left | WALK LTURN RUN |
| look left twice and run opposite right | LTURN LOOK LTURN LOOK RTURN RTURN RUN |
| **jump** twice before walk | JUMP JUMP WALK |
| turn right after **jump** twice | JUMP JUMP RTURN |
| **jump** left twice after **jump** right | RTURN JUMP LTURN JUMP LTURN JUMP |

Table 1: SCAN input commands (left) and output action sequences (right) for Jump task. Upper section is for training, and lower section is for testing. In training, "jump" only appears as a single command. In test, it appears with other words.

We have two different settings in the experiment. In the first setting, we have the following data distribution. In training, the data are generated from the original training dataset. The samples are chosen uniformly at random in the corresponding conditions. $Y_1$ is chosen from all possible labels, and $X_1$ with the label is chosen. $Y_2$ is chosen from $\{Y, Y+1\}$ (we use modular for labels), and $X_2$ with the label is chosen. In test, the data are generated from the original test dataset. $Y_1, X_1$ are chosen in the same way as in training, but $Y_2$ is chosen from the other eight classes $\{Y+2, Y+3, \ldots, Y+9\}$, and then $X_2$ with the label is chosen. This means, in training, $X_2$ contains a part of information for $Y$, and $X_1$ contain all information for $Y$. This is because $Y_2$ is $Y$ with half chance, and $Y_1$ is always $Y$. We hope the model is trained to make $Y$ not dependent on $X_2$.

The second setting has the same test distribution as the first setting, but the training distributions are different. In training of the second setting, $Y_2$ is chosen from all possible labels, so that $X_2$ does not have information for $Y$.

For both the first and second settings, we use a standard convolutional neural network model with three convolution layers and two fully-connected layers. The details of model design and optimization can be found in Appendix A.

The results are shown in Figure 3a. The training accuracy improves faster in the first setting than in the second one, indicating that $X_2$ helps to train the model quickly. In the first setting, the gap of training and test accuracy is significantly larger than that in the second setting, meaning the model does not learn compositionality in the first setting. Therefore, this experiment shows that the redundant information $X_2$ is used to help training model quickly, but the model does not learn compositionality.

## 5.2 LANGUAGE LEARNING

We also run an experiment of instruction language learning with SCAN dataset (Lake & Baroni, 2018). We focus on Jump task, the most difficult task in the dataset. The input is a command instruction, and the output is a corresponding action sequence. The training data include a one-word command "jump", but other training data do not contain the word. In test data, the word "jump" appears in a multiple-words sentence with other words. Please see Table 1 for examples.

In this task, syntax and semantics are two entangled components, and it requires compositional generalization to new combinations. Syntax is the way the actions are organized, and semantic is the mapping from word to action. We also design two settings in this experiment. In the first setting, we use the original training and test data, which are from different distributions. In the second setting, we remove the dependency of input semantics to output syntax by using action words uniformly at random, but we still keep the correspondence between input words and output actions.

We use a standard sequence-to-sequence model with LSTM and attention. More details can be found in Appendix B. Following previous works on SCAN dataset, we use sentence accuracy as the evaluation metric. The results are shown in Figure 3b. We can observe that the training accuracy increases faster during the early training (steps 10-25) in the first setting than in the second setting. Also, the gap of training and test accuracy is significantly larger in the first setting than in the second setting, so the model does not learn compositionality in the first setting.

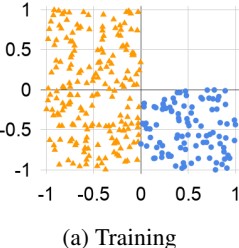 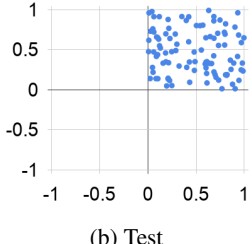

(a) Training          (b) Test

Figure 4: Data distribution for binary classification problem for $Y_1$ output. Horizontal axis is $X_1$ and vertical axis is $X_2$. Blue circle points are positive samples ($Y_1 = 1$). Orange triangle points are negative samples ($Y_1 = 0$).

## 6   DISCUSSION

We use a simple case to visualize the process for better understanding. We consider two random variables $Y_1, Y_2 \in \{0, 1\}$. In training, their joint distribution is uniform on combinations $(Y_1, Y_2) \in \{(0,0), (0,1), (1,0)\}$, and zero on $(1,1)$. In test, $(Y_1, Y_2) = (1,1)$. We use uniform noises with range of 1, by subtracting them from the values (Figure 4).

$$X_1 = Y_1 - U[0,1] \qquad\qquad X_2 = Y_2 - U[0,1]$$

We study a prediction problem with two inputs $X_1, X_2$ and one output $Y_1$. This problem has a property that $X_1$ contains entire information of $Y_1$, but $X_2$ only contains part of information of $Y_1$. $Y_1$ can be predicted from $X_1$ alone, and $X_2$ is redundant for $Y_1$. We want to train a model $f$ with parameters $\theta$. $f$ has $X_1, X_2$ as input and $\hat{Y}$ as output: $\hat{Y} = f(X_1, X_2; \theta)$.

This problem requires compositional generalization, and the model needs to have compositionality. The model has three fully-connected hidden layers with ReLU activations, and each hidden layer has eight nodes. More details can be found in Appendix C.

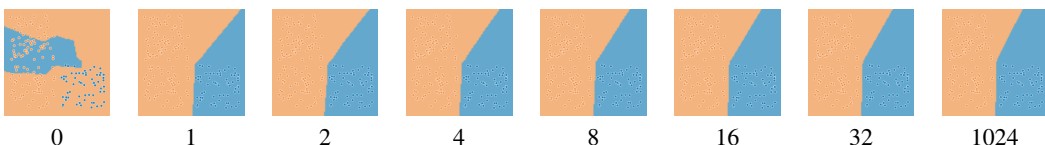

Figure 5: Change of decision boundary for each training step in a binary classification task. In the first training step, $X_2$ (vertical) is helpful for training (step 0), so that the model is updated to cover a part of upper right region as negative (step 1). In the following steps, the loss signals do not completely remove the negative cover in this region, so that the influence remains in the trained model.

Figure 5 shows the decision boundary for each training step. We see that the initialized parameters at step 0 output wrong predictions for samples in the upper left area, so that the samples in this area are useful to reduce loss, and they push the updated boundary in step 1 to the middle of upper right area. In the following steps, the loss is low and do not change much, so that the boundary remains stable. Therefore, the trained model does not have good compositionality. Note that there can be different solutions for this problem. We use different random seeds, and show that the trained model do not have compositionality (Figure 6).

This visualization shows an example of process for how the gradient descent makes the model to be non-compositional.

## 7   CONCLUSIONS

In this paper, we investigate why standard neural network training seldom achieves compositional generalization by studying the relation between compositionality learning and gradient descent during

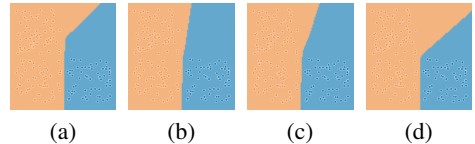

Figure 6: Decision boundaries after 1024 training steps with different random seeds.

training. We find that the optimization process poses a bias towards non-compositional solutions, and this is caused by gradient descent. It tends to use all possible and redundant information from input, so that it violates conditional independence property of compositionality. Based on this study, we suggest that if only model structure design is considered in compositionality learning, it is hard to achieve good compositionality. We hope this finding provides new understanding of compositional generalization mechanisms and helps to improve machine learning algorithms for higher level of artificial intelligence.

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

## A   IMAGE CLASSIFICATION

The model is a convolutional neural network with the following layers. The input is a gray scale image with width 56 (two times of the original width) and height 28. The first hidden layer is a two dimensional convolutional layer with kernel size $3 \times 3$, depth 32, and ReLU activation. The second hidden layer is a two dimensional max pooling layer with kernel size $2 \times 2$. The third hidden layer is a two dimensional convolutional layer with kernel size $3 \times 3$, depth 64 and ReLU activation. Then the representation is flatten to a one vector. The fourth hidden layer is fully connected layer with 64 nodes and ReLU activation. The Fifth hidden layer is another fully connected layer with 64 nodes and ReLU activation. The output layer is a fully connected layer with 10 nodes and linear activation.

We use mini-batch size of 64, and we run 200 steps in training. We use hinge loss, Adam optimizer (Kingma & Ba, 2014) with learning rate of 0.001. We use TensorFlow (Abadi et al., 2015) for implementation.

## B   LANGUAGE LEARNING

We use a standard sequence to sequence architecture. It has embedding layer, bidirectional LSTM encoder, and unidirectional LSTM with attention decoder. The first and last states of encoder are concatenated as initial state of decoder. The embedding size is 32. The state size is 16 for encoder, and 64 for decoder. We run experiment on Jump task in SCAN dataset. The task contains 14,670 training and 7,706 test samples. In the first setting, we use the original dataset. In the second setting, we changed 'jump' to other action word and corresponding output symbol uniformly at random.

We use Adam for optimization, with cross entropy loss. We ran 2,000 training steps. Each step has a mini-batch of 256 samples randomly and uniformly selected from training data with replacement. Initial learning rate is 0.01 and it exponentially decays by a factor of 0.96 every 100 steps. We use TensorFlow for implementation.

## C   VISUALIZATION

The model is a fully-connected neural network, with two input nodes and two output nodes. It has three hidden layers with ReLU activations, and each hidden layer has eight nodes. We use mini-batch size of 10, and we run 1024 steps in training with learning rate 0.1. Please see the original work (Smilkov & Carter, 2016) for more information.

