# OpenReview forum: "Gradient Descent Resists Compositionality"
_ICLR.cc/2021/Conference — Reject_

### Official Review · AnonReviewer3 · 2020-10-28
**This paper studies the relationship between gradient descent and the compositionality generalization, but the result seems to be wrong.**

**Rating:** 3
**Confidence:** 3

**Review:**

This paper studies if gradient descent will affect the compositionality generalization. It attempts to prove the results by information theory and demonstrated several experimental results. Unfortunately, I think the proof has mistakes, and the conclusion doesn't hold.

The major claim of the paper is that the gradient descent tries to use all available and redundant information from input. This is not true. Let's assume that we have two input $x_0$, and $x_1$, and the ground truth is the identity function $f(x_0, x_1)=(x_0,x_1)=(y_0,y_1)$. Now we assume the neural network be a simple linear transformation $f_{\phi}(x_0,x_1)=(a_0x_0+a_1x_1,b_0x_0+b_1x_1)$, and we initialize it with the optimal solution $a_0=b_1=1,a_1=b_0=0$. The partial gradient is then $\frac{\partial y_0}{\partial x_0}=\frac{\partial y_1}{\partial x_1}=1; \frac{\partial y_1}{\partial x_0}=\frac{\partial y_0}{\partial x_1}=0$. The gradient descent will not use any redundant information. As for random initialization case, simple experiments show that the neural network can learn the identity mapping with enough data under the MSE loss (this is obvious as linear regression is a convex problem..). The neural network will have good compositional generalization.

The mistake might be in section 4.3. In the proof, the authors assume the output $\hat{Y}_j$ for any $j\neq i$ is fixed given that $X_j$ has information to reduce the loss. However. if $X_j$ correlates with $\hat{Y}_j$, one can't fix $\hat{Y}_j$ as it may change according to $X_j$. $X_j$ can reduce the loss $\mathcal{L}(Y,\hat{Y})$ by changing $\hat{Y}_j$, then $\frac{\partial \hat{Y}_i}{\partial \hat{X}_j}$ can be zero.

So, I think it's a clear rejection.

---

> ### Author Response · Authors · 2020-11-16
> **Reply to Reviewer 3**
>
> Thank you for the comments.
>
> Q1: The major claim of the paper is that the gradient descent tries to use all available and redundant information from input. This is not true when initializing with the optimal solution.
>
> A1: This paper says gradient descent introduces a bias during the optimization process. In this case, the initialization already achieves the solution without optimization.
>
> Q2: As for random initialization case, simple experiments show that the neural network can learn the identity mapping with enough data under the MSE loss (this is obvious as linear regression is a convex problem..). The neural network will have good compositional generalization.
>
> A2: For a convex problem with only one possible solution, the bias during optimization does not show effect, because anyway that is the only solution.
>
> Q3: The mistake might be in section 4.3.
>
> A3: As discussed above, this is not a mistake.

---

> > ### Comment · AnonReviewer3 · 2020-11-16
> > **Replay**
> >
> > Hi, thanks for the reply. Can you provide me a toy example of how gradient descent bias towards non-compositional solutions? I don't understand what you are talking about.
> >
> > BTW, I don't think the discussion above explains what I say about the mistake in 4.3. Can you update the paper so that the proof is easier to follow?

---

> > > ### Author Response · Authors · 2020-11-23
> > > **Reply to Reviewer 3**
> > >
> > > Thank you for your further questions.
> > >
> > > Q1: Can you provide me a toy example of how gradient descent bias towards non-compositional solutions?
> > >
> > > A1: The example in the discussion section is a visualized example for how gradient descent bias towards non-compositional solutions.
> > >
> > > Q2: I don't think the discussion above explains what I say about the mistake in 4.3. Can you update the paper so that the proof is easier to follow?
> > >
> > > A2: We updated Section 4.3 with more details.

---

### Official Review · AnonReviewer4 · 2020-10-28
**Not sure the paper is about compositionality, it look more like invariances**

**Rating:** 4
**Confidence:** 3

**Review:**

Summary: the paper investigates what neural networks learn when trained with gradient descent, in case parts of the inputs are only partially relevant to the output. The main claim is that GD is what prevents compositionality. In a set of synthetic experiments it is shown that indeed GD learns to use all information in the input, which results in poor generalization ood when only a subset of it was relevant.

My main concerns are the following:

1) It seems to me that compositionality is not really the main aspect of the paper and it is not being tested.
The examples that the authors make earlier in the paper (e.g. shape and colour being entangled in the image), do not reflect the data used later in the experiments. For example, in the case of MNIST there seems to be only 1 factor that is relevant (the digit on the left side), and 1 factor that is spurious (the digit on the right). The test data does not require any compositionality to be solved, but only finding the invariance instead.
So my impression is that the paper in its current form is much more about finding invariances than learning compositionality. Nothing needs to be combined in order to solve the test set, the invariant mechanism is what’s needed.

2) The paper is purely of “descriptive” nature, i.e. not “prescriptive” at all. While describing a novel problem can be already sufficiently interesting in general, I am under the impression that the general problem of learning invariances in neural nets as described in the paper has already been identified before [e.g., Heinze-Deml et al., 2018; Arjovsky et al., 2019].
Most papers on the topic are now of a prescriptive nature, in the sense that they also investigate potential solutions to the problem.



Minor:
- By the time Figure 1 is mentioned, the caption mentioned “entangled”, but it’s unclear what that means in this context (and as a consequence it’s hard to interpret the figure).

- P6: “Y2 is chosen from {Y, Y + 1}” : do you mean $Y_i$ for both instead of $Y$? (And again later two lines below?)

(Not affecting the score, but the paper should be carefully proof-read for English syntax before publication.)

C. Heinze-Deml, J. Peters, and N. Meinshausen. Invariant causal prediction for nonlinear models. Journal of Causal Inference, 6(2), 2018.

M. Arjovsky, L. Bottou, I. Gulrajani, and D. Lopez-Paz. Invariant risk minimization. arXiv preprint arXiv:1907.02893, 2019.

---

> ### Author Response · Authors · 2020-11-16
> **Reply to Reviewer 4**
>
> Thank you for your feedback.
>
> Q1: It seems to me that compositionality is not really the main aspect of the paper and it is not being tested.
>
> A1: This paper shows that conditional independence property is a key to compositional generalization, and gradient descent breaks the property so that it does not support compositional generalization. The experiments show examples that break the property, as stated in the paper.
>
> Q2: The general problem of learning invariances in neural nets as described in the paper has already been identified.
>
> A2: This paper is not about learning invariance, as discussed in Q1.
>
> Q3: By the time Figure 1 is mentioned, the caption mentioned “entangled”, but it’s unclear what that means in this context.
>
> A3: Thank you. We will clarify the context in the caption.
>
> Q4: P6: “Y2 is chosen from {Y, Y + 1}” : do you mean for both instead of? (And again later two lines below?)
>
> A4: This means Y2 is one of Y or Y+1. The later one is the same.
>
> Q5: (Not affecting the score, but the paper should be carefully proof-read for English syntax before publication.)
>
> A5: Thank you. We will revise carefully.

---

> > ### Comment · AnonReviewer4 · 2020-11-22
> > **Reply to reply**
> >
> > _A1: This paper shows that conditional independence property is a key to compositional generalization, and gradient descent breaks the property so that it does not support compositional generalization. The experiments show examples that break the property, as stated in the paper._
> >
> > Your answer to Q1 does not go into any detail at all in trying to answer the concern that the experiment is not about compositional generalization, given that there is only one factor.
> > Can you please elaborate? And then again in Q2?

---

> > > ### Author Response · Authors · 2020-11-23
> > > **Reply to Reviewer 4**
> > >
> > > Thank you for the question.
> > >
> > > A: This paper studies compositional generalization by studying conditional independence property (it is not invariance), which is a key property for the generalization. Please see Section 3 for more details. The experiment is designed to show the effect of breaking the conditional independence property, which focuses on one output factor and multiple input factors.

---

### Official Review · AnonReviewer2 · 2020-11-01
**Initial Review**

**Rating:** 1
**Confidence:** 4

**Review:**

review:
This paper addresses the effects of gradient descent methods onto compositionality and compositional generalization of models. The authors claim that the optimization process imposes the models to deviate compositionality, which is defined with conditional independence among random variables of input, predicted output and the ground-truth. Since compositionality is one of important features of human intelligence, it has been interested widely in the field of AI/ML such as vision, language, neuro-symbolic approaches, common sense reasoning, disentangled representation, and the emergence conditions of compositionality. As it has been not much focused on the relationship with optimizers, it is fresh and interesting. However, it is not easy to figure out the position of this paper from two reasons: (1) the definitions on compositionality in this paper are not so compatible with recent related works, which mostly consider certain structures in models [ICLR19, JAIR20] or representative problems such as visual reasoning [CVPR17] and Raven progressive matrices [PNAS17]. (2) The authors do not consider quantitative approaches such as compositionality [ICLR19] or compositional generalization [ICLR20].

In this paper, the main claim is very broad argument. To verify this claim, the authors provide supports of both theoretical and experimental aspects. Theoretically, they try to show that reducing loss values in the optimization process induces utilizing other input variables including useful information based on mutual information. Experimentally, they show the gaps between several settings of accuracy curves with the MNIST dataset (vision) and the SCAN dataset (language). With both aspects, theoretical steps are vague and weak, and the experimental results are little persuasive and convincing.
Some steps in theoretical derivation seem to be wrong.
I recommend ‘trivial and wrong’ for this paper.

Pros:
They deal with the relationship among compositionality, compositional generalization and gradient descent. It is interesting and novel question as far as I know.

Concerns:
-	It is not clear the assumptions on models is covered in the main claim. Some arguments have readers guess the claim only on neural networks. Currently, it is not explicit. What if a model is naïve Bayes classifier which assumes conditional independence? Does it have compositional generalization? If the classifier is trained with gradient descent, the key argument of the paper has counterexamples, which becomes wrong.
-	Theorem 1 should show more clearly Markov chain structure among X, Y and Z. X -> Y -> Z (as written in Cover 1999 p.34)
-	What is the relationship between Y and X in Proposition 1?
-	The proof in Proposition 2 seems not valid. Is the Markov chain among Y hat, X, and Y still valid? Without any constraints of X and Y, the equation in the middle of Proposition 2 seems not an identity (consider joint probability models with discrete values), and the derivation process is not trivial. The validity of this result is a factor that also affects subsequent verification.
-	There is no quantitative analysis with measurable cases as mentioned above.


[CVPR17] Johnson et al., CLEVR: A diagnostic dataset for compositional language and elementary visual reasoning, CVPR 2017.

[PNAS17] Duncan et al., Complexity and compositionality in fluid intelligence, PNAS 2018.

[ICLR19] Jacob Andreas, Measuring compositionality in representation learning, ICLR 2019.

[ICLR20] Keysers et al., Measuring compositional generalization: a comprehensive method on realistic data, ICLR 2020.

[JAIR20] Hupkes et al., Compositionality decomposed: how do neural networks generalise?], JAIR 2020.

---

> ### Author Response · Authors · 2020-11-11
> **Reply to Reviewer 2**
>
> Thanks for considering that this paper addresses an interesting and novel question.
>
> Q1: The definitions on compositionality in this paper are not so compatible with recent related works, which mostly consider certain structures in models [ICLR19, JAIR20] or representative problems such as visual reasoning [CVPR17] and Raven progressive matrices [PNAS17].
>
> A1: The definition in this paper follows previous related papers, such as [ICML18] and [ICLR20b].
> [ICML18] Lake et al. Generalization without systematicity: On the compositional skills
> of sequence-to-sequence recurrent networks
> [ICLR20b] Li et al. Compositional Language Continual Learning
>
> Q2: The authors do not consider quantitative approaches such as compositionality [ICLR19] or compositional generalization [ICLR20].
>
> A2: This is a theoretical paper. We do not use experiments to prove the claim. The illustrative examples are used to help understand the theoretical results. We quantitatively measure the ability of compositional generalization, and we consider compositionality from the aspect of being able to achieve compositional generalization.
>
> Q3: It is not clear the assumptions on models is covered in the main claim. Some arguments have readers guess the claim only on neural networks. Currently, it is not explicit. What if a model is naïve Bayes classifier which assumes conditional independence? Does it have compositional generalization? If the classifier is trained with gradient descent, the key argument of the paper has counterexamples, which becomes wrong.
>
> A3: We focus on neural networks.
>
> Q4: Theorem 1 should show more clearly Markov chain structure among X, Y and Z. X -> Y -> Z (as written in Cover 1999 p.34)
>
> A4: Thanks for pointing. We will make it clear that “X->Y->Z is a Markov chain” in Theorem 1.
>
> Q5: What is the relationship between Y and X in Proposition 1?
>
> A5: X is input and Y is output of a neural network.
>
> Q6: The proof in Proposition 2 seems not valid. Is the Markov chain among Y hat, X, and Y still valid? Without any constraints of X and Y, the equation in the middle of Proposition 2 seems not an identity (consider joint probability models with discrete values), and the derivation process is not trivial. The validity of this result is a factor that also affects subsequent verification.
>
> A6: The Markov chain is valid, as explained in the first sentence of the last paragraph before Proposition 2. “In the common supervised learning setting, given X, the ground truth Y does not depend on prediction Y hat”.
>
> Q7: There is no quantitative analysis with measurable cases as mentioned above.
>
> A7: See A2.

---

> > ### Comment · AnonReviewer2 · 2020-11-17
> > **Reply for the answer**
> >
> > Hello, thank you for the reply.
> >
> > Could you explain more about the equation in Proposition 2 and the derivation as I mentioned in Q6?
> > A6 does not consider the part which it seems it is crucial.

---

> > > ### Author Response · Authors · 2020-11-23
> > > **Reply to Reviewer 2**
> > >
> > > Thank you for asking the question.
> > >
> > > A: We use linear approximation so that dx = Δx. By definition, Δf(x) = f(x+Δx) - f(x).
> > > We updated the paper, and used Δ to make it more clear.

---

### Official Review · AnonReviewer1 · 2020-11-02
**interesting perspective on compositionality, but the claim seems too strong and not supported by experiments**

**Rating:** 5
**Confidence:** 3

**Review:**

This work analyzes the effect of gradient descent training on the compositionality of the learned model. It is shown that the gradient descent would use all the available information, even when it is redundant to learn the mapping from input to the output. It is then argued that the gradient descent training has the bias against compostionality despite the model architecture. Experiments are conducted on three simple benchmarks to demonstrate that when gradient descent trained model would use redundant information and not generalize compositionally.

Strength:

1. This work takes a new perspective to analyze the lack of compostionality in the neural network models and focuses on how gradient descent training violates the conditional independence of the inputs.

Weakness:

1.  The theorem doesn't seem to add too much new information since the conclusion that gradient descent leverages redundant information seems quite straightforward given it is taking the partial derivative w.r.t to each input. It would perhaps help to highlight the *new* insights from the proofs.

2. It is not convincing that the gradient descent alone is the sufficient reason for violating conditional independence and causing lack of compositionality. As a simple example, given input with feature X1 and another redundant feature X2, if the model is linear with sparsity encouraging regularization such as L1, the model would probably learn to use only X1 even when trained with gradient descent. The experiments do show that the trained model is not able to neglect the redundant information, and the redundant information makes the training faster, but it doesn't support the claim that model architecture design couldn't help or achieve compostionality if trained by gradient descent. In fact, there are recent works trained by gradient descent that achieved perfect generalization on datasets like SCAN (Chen et al, 2020) by leveraging a better model design.

Chen, Xinyun, et al. "Compositional Generalization via Neural-Symbolic Stack Machines." arXiv preprint arXiv:2008.06662 (2020).

---

> ### Author Response · Authors · 2020-11-16
> **Reply to Reviewer 1**
>
> Thank you for the comments.
>
> Q1. The conclusion seems quite straightforward. Please highlight the new insights from the proofs.
>
> A1: It has not been clear why many standard neural networks seldom achieve compositional generalization. We like to highlight that a key point is the connection between gradient to compositional generalization.
>
> Q2. It is not convincing that the gradient descent alone is the sufficient reason for violating conditional independence and causing lack of compositionality.
>
> A2: Gradient introduces the bias towards non-compositional solutions during optimization. For a linear model with only one solution, the bias does not have effect.
>
> Q3. In fact, there are recent works trained by gradient descent that achieved perfect generalization on datasets like SCAN (Chen et al, 2020) by leveraging a better model design.
>
> A3: The pointed paper also uses methods other than architecture design, e.g. curriculum learning and execution trace searching. So it is a good example that architecture design alone is not enough.

---

### Decision · Program_Chairs · 2021-01-07
**Final Decision**

**Decision:**

Reject

**Comment:**

Dear Authors,

Thank you very much for submitting this very interesting paper.

This work analyzes the effect of gradient descent training on the compositionality of the learned model. Their main argument is that GD tries to use the redundant information in the data and, as a result, it doesn't generalize well. The paper then tries to show that theoretically and empirically with some simple experiments.

There is a general consensus among all the reviewers that this paper is not suitable for publication at ICLR. The authors do not entirely address most of the concerns raised by the reviewers during the rebuttal.

If the authors improve the clarity of the paper, making some of the propositions and theories more concrete and grounded in experiments as well, I would recommend them to resubmit this paper to a different venue since the premise of the paper is important and interesting.

Some of the reasons:

- The paper claims that the gradient descent can not ignore the redundant information without providing sufficient empirical results. Though the part that is not clear to me whether if it is a credit assignment or an optimization problem. I agree with R1 that it is not clear what type of new insights from the proofs.

- As R1 mentions, this paper's claim seems too strong and not supported by experiments.

- R2 finds part of the paper unclear and thinks that some of the paper's propositions and theories are either trivial or wrong. The rebuttal doesn't seem to be doing a good job in terms of addressing those concerns.

- R4 also is confused with the paper thinks that some of the theories are incorrect.